# Strengthening Partnerships to Safeguard the Future of Herbaria

Barbara M. Thiers [1,2]

1    New York Botanical Garden, Bronx, NY 10458, USA; bthiers@nybg.org
2    Denver Botanic Gardens, 1007 York St., Denver, CO 80206, USA

**Abstract:** Herbaria remain the primary means of documenting plant life on earth, and the number of herbaria worldwide and the number of specimens they hold continues to grow. Digitization of herbarium specimens, though far from complete, has increased the discoverability of herbarium holdings and has increased the range of studies from which data from herbarium specimens can be used. The rather large number of herbaria about which no current information is available is a source of concern, as is herbarium consolidation and removal of herbaria to offsite storage facilities. Partnerships are key to the future health of herbaria. Benefits could accrue from the reimagining of the world's herbaria as a global resource rather than a collection of independent, often competing institutions. Herbaria can extend the reach of their specimens by joining the nascent effort to link the species occurrence data they manage to other biological and environmental data sources to deepen our ability to understand the interrelationships of earth's biota. To assure that data held by herbaria contribute to the range of conservation-related projects for which they are relevant, herbaria should embrace the tenets of Team Science and play a more proactive role in promoting their holdings for relevant research and conservation projects.

**Keywords:** biological collections; herbaria; digitization; historical collections; museological importance; vascular plants; algae; bryophtyes; fungi; lichens

## 1. Introduction: The Current State of the World's Herbaria

An invention of Renaissance Europe, the herbarium has endured for five centuries as the primary means of documenting plant life and has spread to all corners of the earth. The most objective source of information on the current state of the world's herbarium is Index Herbariorum [1], which compiles information, provided by herbaria themselves, about the location, holdings and staff for each herbarium that chooses to register itself in the index. In this article, the term "herbarium" includes collections of fungi, which are sometimes referred to as fungaria, as well as algae, bryophytes, and lichens. Although Index Herbariorum began in the 1930s, a snapshot of the data has only been captured on an annual basis since 2016. According to the most recent annual report [2], as of 31 December 2022 there are 3567 active herbaria in the world, containing 396,746,986 specimens. Associated with the world's herbaria are 13,717 staff members and associates. Ninety three percent of the world's countries have at least one herbarium. Since 2016, the number of registered herbaria has increased by 605, and 92 herbaria have been reported as inactive. During this period, the number of specimens reported by the world's herbaria has increased by more than 15 million specimens and the number of associated staff by approximately 2100.

Positive trends for herbaria. Beginning in the 1990s, but increasing in intensity from about 2010 until today, herbarium specimen digitization has provided one of the most significant advances in the history of herbaria [3–5]. Although digitization originally referred only to the transcription of specimen label data into a structured database, now digitization routinely involves capturing a digital image of the specimen as well, a procedure for which, due to their flat shape, herbarium specimens are particularly well suited.

Digitized herbarium specimen data are aggregated and shared through several data portals. These portals may be international, such as the Global Biodiversity Information

Facility [GBIF] or iDigBio data portals [6,7]; geographically based, such as those for herbarium specimens from Australia [8], Brazil [9], China [10] Colombia [11] and Mexico [12]; or taxon-based portals for fungi [13], ferns and related groups [14], and fungi [15]. Often, datasets in the "specialty" portals are also served to GBIF and iDigBio. The number of herbarium specimens available online through GBIF as of 14 November 2023 is 231,562,216, constituting 58% of the specimen total reported in Index Herbariorum.

Digitization and the open sharing of data on worldwide data portals has made it possible for herbaria to greatly extend their reach, and it has been an especially important way for researchers in formerly colonized countries to gain access to specimens that were collected during colonial-era expeditions [16]. The new research uses of herbarium specimens to gain a deeper understanding of plant biodiversity are ably summarized by several authors [17–20]. Herbarium specimen data have helped to identify the species most at risk of extinction [21–23], why species become invasive [24], and how relationships between plants and the animals that pollinate and predate them have changed over time and may change in the future [25,26]. Curricula and lesson plans compiled through projects such as Biodiversity Literacy for Undergraduate Education [27] demonstrate how digitized herbarium specimens can be used to teach data literacy.

*Negative trends for herbaria.* Despite the success of herbarium digitization, the perception remains that the herbarium enterprise is in peril, as posited in articles such as that of Deng [28]. The herbarium decline cited in this article was evidenced by several highly publicized herbarium closures in the U.S., i.e., the University of Missouri at Columbia and the Brooklyn Botanical Garden in New York City in 2015 [29,30], and the University of Louisiana at Monroe in 2017 [31]. In all these cases, other herbaria absorbed the specimens from those that closed, so there was no loss of specimens.

Although worldwide decline in herbaria is not supported by a decrease in the number of active herbaria, there are many herbaria for which we have no recent data. Despite repeated efforts to encourage them to do so, nearly 800 herbaria have not updated their contact or other information for Index Herbariorum in more than 15 years. It is possible that some of these herbaria no longer exist, or have no one actively managing the collection, and thus the collections they hold (approximately 25 million specimens in total) may be lost or in peril. A list of the herbaria in this category can be found in the Supplementary Material.

In recent years, several prominent institutions have made the decision to move their herbarium from a location central to their administrative functions to a more remote location. The University of Michigan Herbarium (Index Herbariorum code MICH) was moved from the central campus to a location seven km from central campus in 2002. In 2022, the Sydney Botanic Gardens herbarium (NSW) was moved from the main grounds of the Royal Botanic Gardens Sydney near Sydney Harbor to Mt. Annan, New South Wales, 59 km away. In 2023, the Royal Botanic Gardens, Kew, announced plans to move the herbarium (K), the second largest in the world, to a new facility at the Thames Valley Science Park in Reading, approximately 57 km from the Botanic Gardens' main grounds in Richmond. In the case of MICH and NSW, the new quarters provided ample well-configured space for maintaining current herbarium functions, allowing for the possibility of expansion of those functions. Plans announced for the new Kew herbarium facility suggest that the new facility would likewise be designed to enhance and expand current herbarium functions [32]. However, relocating a collection from a prominent location within an institution to a remote one causes fear that the herbarium will become marginalized administratively (especially budgetarily) as well as physically. Removing the herbarium from the center of the institution is seen as symbolic of a belief by the institution's leadership that herbarium is not central to the institution's mission. Such a move may also complicate work travel for staff and in-person visits to the collection [33,34]. However, in many cases, the herbarium's facilities may be significantly upgraded through the move and the institution can point to the major new financial investment for the new facility as a commitment to the value of the collection rather than an indication of marginalization. Given that most of these relocations are relatively recent (or in the case of Kew, only planned), it is not possible yet

to make any objective measure of impact of relocation on the ability of the herbarium to fulfill its mission.

*Challenges facing herbaria in the 21st century.* Herbaria hold data that are key to understanding how earth's vegetation has changed over time, and how it may change in the future, an issue of critical importance for everyone on the planet. Given the urgency of these issues, it can be argued that herbaria have never been more relevant for humanity. Whether or not the herbarium enterprise will grow and flourish over the next century or be increasingly marginalized from mainstream science will depend on how active a role those responsible for the care of the world's herbaria are willing to play in demonstrating that relevance. The conundrum of how to take on additional roles at a time when some herbaria resources are in decline and existing staff are already stretched to the limit may be solved, at least in part, by the creation of new partnerships or reimaging of existing ones, and perhaps adjusting priorities and former measures of success.

## 2. New Initiatives for Herbaria

### 2.1. One Herbarium

Recent articles have suggested that natural history collections should think of themselves less as stand-alone entities and more as components of a global resource whose decisions about research and collection foci are guided by community-wide principles and priorities. Johnson & Owen, along with nearly 150 coauthors [35], urge consideration of the collective power of global natural history collections and suggest that "strategic coordination and use of the global collection has the potential to focus future collecting and guide decisions that are relevant to the future of humanity and biodiversity". The concept of an Open-Specimen movement was proposed by Colella et al. [36] as "a paradigm shift from specimen ownership to specimen stewardship". If herbaria were to change their self-perception to one of stewards of an international resource that exists for the good of all people rather than a set of independently owned, sometimes competing collections, such a partnership would open the door to changes that could potentially strengthen all herbaria.

The herbarium community already has a reputation among natural history collections as being particularly cohesive, as evidenced by their participation in Index Herbariorum, their extensive exchange and gift for determination networks [37], and their digitized specimen consortia, including the Global Plants Initiative database of type specimens [38], a collaboration of 270 herbaria in more than 70 countries. More than many other biological data communities, herbaria have embraced the concept of FAIR (Free, Accessible, Interoperable and Reusable) data [39]. The digitized data that herbaria share through the various data portals have already been characterized by Davis [40] as "an open-access global meta-herbarium", and thus the One Herbarium concept is simply of an extension of that concept to physical specimens as well. The community is therefore already somewhat primed to adopt a One Herbarium mindset, and to work collectively to articulate our common values, mission, and goals, and set priorities for future growth and research, while of course individually continuing to pursue the research project to which they are best suited.

### 2.2. Advantages of a One Herbarium Concept

2.2.1. Research and Guidance for Future Planning

Pursuing the goal of uniting the world's herbaria into a global herbarium would allow the community to articulate common values, mission, and goals, determine collection gaps, and set priorities for future collecting. Together, we might gain clarity on questions that are fundamental to herbarium accession policies, such as how many specimens and over what time and geographic ranges are needed to determine the range of a species [41], and to track changes in that distribution range over time. Fruitful discussions in zoological collections have yielded rubrics for collection evaluation that can be applied across collections, and these may be adaptable for botanical collections as well [42–45]. For herbarium specimen data to reach their maximum potential for contributing to efforts to monitor and manage natural capital [46], we need to identify and collectively fill the gaps in our knowledge of the

biodiversity in particular ecosystems. We need to understand the extent to which herbarium specimens, perhaps in combination with observational records, can be used to provide "real time" approximations of species distribution, biomass, and biotic relationships.

### 2.2.2. Greater Equity in Plant Diversity Science

There have been many recent calls for increasing diversity, equity, and inclusion in science, so that benefits of new technologies can be accessible to all people, and all can contribute to addressing environmental challenges [47,48]. This process is sometimes referred to as decolonizing science, alluding to the continued dominance of current and former colonial powers in research. Adoption of a One Herbarium concept could open the door to decolonization of our collections. Park et al. [49] demonstrate that plant diversity as represented by herbarium specimens is greatest in Europe and North America, although the floristic diversity of these regions is far less than that in many countries in the Global South. In other words, there is an inverse relationship between living plant diversity and the preserved representatives of that diversity held in herbaria. Although digitization has helped to increase access to the biodiversity held in northern herbaria for scientists in the south, the absence of physical specimens may limit the ease with which existing specimens can be used for studies that require access to the actual specimen (e.g., DNA or micromorphological analyses). Herbarium traditions allow for the loaning of specimens, although some herbaria can neither send or receive loaned specimens due to costs, or postal or quarantine requirements.

Repatriation of museum specimens is the focus of a great deal of current discussion in museums [50,51]. As demonstrated by Park et al., the same colonial impulses that led Europeans and North Americans to remove cultural artifacts from their home countries to be housed at museums in colonizing countries drove accessions to northern herbaria as well. A major difference between herbarium specimens and cultural artifacts, and indeed many other natural history collections as well, is that herbarium specimens are often collected in sets of duplicates; that is, multiple sheets of specimens gathered from the same plant at the same time. Today, international specimen collection is usually governed by Access and Benefit Sharing agreements as part of the Nagoya Protocol [52], and the requirement for the deposition of the first set of any botanical collections is usually stipulated in the Prior Informed Consent agreements that authorize such collection. However, for most of herbarium history, there have been no restrictions on the removal of plant material from colonized areas. Botanists such as Muthama Muasya of University of Cape Town, South Africa [53], have suggested that such specimens be returned to their country of origin. Although this is a complicated subject that would need to be considered or negotiated on a country-by-country, or perhaps institution-by-institution basis, it is worth acknowledging the potential benefits of specimen repatriation for all parties involved. Many of the largest herbaria in the world are running out of room for future growth, and space restrictions may limit their ability to continue to document earth's biodiversity and engage in other new research projects of interest. One of the main motivations for the proposed relocation of the herbarium of the Royal Botanic Gardens at Kew is a lack of space for future growth of the collection. Repatriating some holdings might free enough space to allow some herbaria to continue to accrue new specimens without facing relegation to offsite facilities.

Repatriation of specimens could also allow larger herbaria to improve curation of their specimens. The Index Herbariorum annual report for 2022 summarizes the number of specimens and the number of herbarium staff in each country. Comparison of the specimens to staff ratio for countries in different regions of the world indicates that Europe and North America have the highest specimens to staff ratio, whereas other regions have much lower ratios (Table 1). There are several caveats for the interpretation of these data. First, the specimen to staff ratio is a crude method for comparing specimen management practices across herbaria because responsibilities and levels of training and time commitment of staff may vary widely among herbaria. Also, the specimen totals used in the computation of the ratios include only accessioned specimen totals, and do not include collections that

may be unidentified or are awaiting processing. Also, herbaria in countries with a long-standing herbarium infrastructure may operate at a higher level of efficiency in specimen management. Nonetheless, it is reasonable to assume that the herbaria with the highest ratios of specimens to staff have less time to care for and study each specimen than those with lower ratios. Identifying the strategic goals for herbaria, on an individual as well as regional, national, and even global basis, would help herbaria to prioritize collection and curation priorities, and focus their efforts on the specimens most relevant to their mission. Elaboration of these goals might be used as a starting point to determine which holdings are most critical for an herbarium to obtain its strategic goals, and which collections might better fit the goals of another herbarium. The effort currently underway to create a data standard for collections description, called the Latimer Core [54,55] and ultimately to link these characterizations through GBIF's Global Registry of Scientific Collections [56] might be a mechanism for determining areas of overlap as well as gaps in herbarium holdings, and in the future might guide limited or large-scale repatriation efforts.

**Table 1.** Regional differences among Herbaria in specimens to staff ratio. The ratio was obtained by dividing the total number of specimens in the herbaria of each region by the total number of associated staff in each region. Combining data for all regions, the average ratio of specimens to staff is approximately 28,000.

| Region | Specimen to Staff Ratio (Rounded to Nearest Whole Number) |
|---|---|
| Europe | 55,310 |
| North America | 35,673 |
| Australasia | 28,907 |
| Middle East and northern Asia | 20,430 |
| Africa | 14,473 |
| Pacifica | 16,086 |
| South and Central America | 9497 |

### 2.3. Extending Herbarium Specimens through Partnership with Other Biological Collections

Opportunities are developing to compound the value of FAIR herbaria data through partnerships with other biological collections and datasets in the development of an extended network of biological data. The extended specimen concept refers to a physical specimen, its digital counterpart, and all of its physical and digital derivatives [57–59]. The study of herbarium specimens has generated a wide range of derivative data elements, such as images (of whole organisms or component parts), microscopic preparations (e.g., pollen grains), wood samples, gene sequences, or other chemical components, as well as documentation of biotic interactions and synthetic research products based on the analysis of multiple specimens such as floras, monographs, and phylogenetic analyses. Through their locality data, herbarium specimens are linked to a wide range of environmental data sources. Thus, the full extension of a specimen includes not just the primary derivative data elements but also the synthetic research projects based on them and any geography-based environmental research. Currently, the elements that may comprise an extended specimen reside in disparate databases that may not be linked directly to the specimens with which they are associated or have not yet been digitized or made accessible to the scientific community. However, through partnerships among the biological data community, it should be possible to bring together vast and currently largely siloed sets of genetic, ecological interaction and environmental data into a network that will allow us to address questions of why plants (and all organisms) live where they do and to predict the impact of their increase or decrease in distribution range due to changing climate conditions.

To create an extended specimen network will require new levels of cooperation among collections on an international level and creating linkages with a broader range of biological and environmental data communities [60]. The extended specimen network concept began in the U.S. but developed in parallel with discussions among European institutions participating in the DISSCo initiative focused on digital specimens [61]. Developers of these

two very similar concepts have now joined forces and expanded the conversation to a wide array of international collaborators under the Digital Extended Specimen initiative [62].

There are considerable technical challenges in the linking of disparate databases and the creation of interfaces that will allow users to fully explore the richness of an extended specimen network. To prioritize the collaborations that would be needed to create and maintain such a network is an even greater challenge, requiring the creation of new partnerships among researchers, bioinformaticians and managers of environmental and public health databases. To begin the conversation, the Biodiversity Collections Network [63] will hold a series of listening sessions and a workshop to explore how a variety of other biological data initiatives might align their priorities to craft a comprehensive network of FAIR biological data. The project, called BIOFAIR, will be carried out between 2024 and 2026, and has financial support from the U.S. National Science Foundation [64]. The proposed outcome of this project will be a roadmap or implementation plan for how to create and sustain a global biological data network that would effectively create a partnership among all biological data providers and users.

A hypothetical example demonstrates how an Extended Specimen Network might link data from herbarium specimens to other relevant data sources for a deeper understanding of differences in the distribution of closely related species. Searching by scientific name, the investigator could access all digitized herbarium specimens of those species and learn of herbaria that might hold relevant but not yet digitized specimens. The responses to this query would also reveal images of specimens, and of ancillary parts, such as pollen grains, or wood anatomy, as well as images of observation-based records for the species. The query would also yield links to taxonomic treatments, gene sequences, phenological data, and information about commensal organisms such as pollinators, predators, and dispersers. Through the locality data of the specimens, the user could access relevant environmental data such as soil types, rainfall, temperature, and other climate data. Currently, all these data could be assembled manually, but the process would be tedious, and the user might well miss some relevant data sources.

A key feature of an Extended Specimen Network will be the ability to track how specimens are used in research. This information will help herbaria comply with the reporting requirements of the Nagoya Protocol and will justify the scientific relevance of their collections. Knowledge of how their specimens are used may drive future research projects as well. For herbaria that have already digitized and shared their specimen data, participation in the Extended Specimen Network as currently envisioned would likely not require more effort for individual herbaria than they already made; linkages among data elements would likely take place through efforts at the major data portals outlined above. However, this initiative could serve as a justification for funding for continued specimen digitization and data editing efforts, and perhaps for the digitization of ancillary collections such as microscope slides, photographs, or collections derived from or related to specimens.

### 2.4. Active Partners in Conservation

The Intergovernmental Science-Policy Platform on Biodiversity and Ecosystem Services (IPBES) issued a report in 2019 [65,66] demonstrating that biodiversity is declining at an unprecedented rate, and that the window for implementing measures to protect our critical biodiversity is closing. Although this 1148-page report includes many suggestions for future actions by a wide number of groups, the role of biological collections is not mentioned. Perhaps this omission stems from ignorance of the resource, or perhaps characteristics of biological collections data limit their use in some studies. These limitations may include data inconsistencies (e.g., misidentifications, multiple names for the same organism), limitations of user interfaces associated with existing search engines, or lack of comparable year-to-year sampling regimes. Whatever the reason, it is the responsibility of herbaria to promote greater use and appreciation of the role of specimen data for conservation research and policy.

The Team Science approach to addressing complex problems suggests a new paradigm for partnerships in which herbaria could serve as active research partners rather than passive data providers [67]. Actively promoting their content for use has not been a traditional role of herbaria. But the imperatives for improving human health and wellbeing that are increasingly driving libraries to bring their content to potential users [68] are equally strong for herbaria and all natural history collections, which, like libraries, obtain public support based on value to humankind. Rather than making data available only when requested, herbaria could suggest sets of data or specimens that are particularly relevant for ongoing studies and promote these actively, just as a library might create and circulate lists of their holdings on topics of current interest to patrons. A full implementation of a Team Science approach would recognize the contributions of collections (both biological and library) to research endeavors and would provide proportional recognition through acknowledgement and compensation.

## 3. Discussion

It will not be easy to achieve consensus about our herbarium community values and priorities at a global, national, or perhaps even a regional level. However, a model exists in the Australasian community with their Council of Heads of Australasian Herbaria and the subsidiary Managers of Australasian Herbarium Collections [69]. These groups, which originally focused only on Australia, were instrumental to the success of the specimen digitization and development of the Atlas of Living Australia data portal, and their experiences could inform the development of similar groups in other areas. Herbaria may be able to build upon existing networks, whether created for sharing digital data or collaboration on floristic projects, to start the discussion of how to achieve consensus on goals and strategies. International professional societies that involve herbarium related activities, such as the Society of Herbarium Curators, the Society for the Preservation of Natural History Collections and both the International Association of Plant Taxonomists and the American Society of Plant Taxonomists could help to provide a forum for these discussions through conferences or publications.

The types of partnerships that might emerge from community-wide discussions about the future of herbaria may be quite different, and indeed far better, than those suggested in this paper. Far less important than any particular initiative would be a recognition of our collective potential and an assertion of our will to see our resource reach its full potential. Institutions such as herbaria or natural history collections in general are often resource-poor to the extent that they do not see a clear path to maintaining the current state of the collection, let alone engaging in new activities. This has led to a "zero sum" mindset, in which collection professionals assume that participation in new initiatives must come at the expense of basic collection care. Recognizing the unrealized potential of herbaria to participate in projects that will help to protect biodiversity and create greater equity among people and communities will hopefully entice herbaria to lean into the spirit of collaboration that has long existed among the world's herbaria, and allow their collective creativity to find ways to continue their basic missions as well as contribute more to global efforts to protect our environment. Among the considerations that must go into such discussions is how to support the herbarium workforce with positions that recognize their contributions with sufficient salaries and reasonable expectations of work deliverables. Again, a collective approach to defining herbarium position requirements and suggesting routes for professional development and advancement within the collections profession could be helpful for individual institutions to offer fairly compensated and intellectually stimulating positions for their employees.

Herbaria, through the geographic and temporal breadth of their holdings and the spirit of cooperation that characterizes the herbarium community, have the potential to model a 21st century action-oriented biological collections community that embraces FAIR principles and achieves equity in partnerships among themselves and with other advocates for the preservation of biological diversity.

**Supplementary Materials:** The following supporting information can be downloaded at: https://www.mdpi.com/article/10.3390/d16010036/s1. Table S1: List of Herbaria of Unknown Status.

**Funding:** This research received no external funding.

**Data Availability Statement:** Publicly available datasets were analyzed in this study. This data can be found here: https://sweetgum.nybg.org/science/ih/.

**Acknowledgments:** I thank the editor of this special issue for inviting me to contribute this paper. I also thank iDigBio for funding my participation in a symposium at the meeting of the Society for the Preservation of Natural History Collections in San Francisco in 2023, where I presented some of the ideas that were developed further for this paper.

**Conflicts of Interest:** The author declares no conflicts of interest.

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
