# Peer review of "Strengthening Partnerships to Safeguard the Future of Herbaria"

_diversity, doi:10.3390/d16010036_

Round 1
Reviewer 1 Report
Comments and Suggestions for Authors
This paper is a very valuable contribution to the topics of this special issue.
I reccomend to publish the manuscript in the present form.
Author Response
No comments made by the reviewer. No response.
Reviewer 2 Report
Comments and Suggestions for Authors
The introduction of the manuscript entitled “Strengthening Partnerships to Safeguard the Future of Herbaria” highlights Herbaria's importance for documenting, understanding, and preserving biodiversity. The author updates the numbers of active and inactive herbaria in the world and herbaria that have not made contact anymore or other information for Index Herbariorum. I confess it would be useful to include these numbers in the world map promoting a critical discussion about the gaps that need to be restored.
The main proposal of the paper is to shed light on new possibilities for herbaria such as “Open-Specimen movement”, “One Herbarium concept”. In these two initiatives, the author could bring examples (fictitious or not) of how it will work in practice.
Concerning the complicated subject of repatriation of the specimens to the country of origin, the author suggests the lack of space for future growth of the collection evidenced by several Herbaria could be a good reason for returning them. During this difficult impasse, it should be discussed which herbarium in the country will receive this historical material. Maybe elect the major Herbarium in the state, region, or province where the plant was collected?
Related to Table 1, Europe and North America have the biggest herbaria in the world (millions of specimens), being most of the specimens already available online in pictures of high resolution. Their infrastructure (physical and digital) is incredible, and the process of curation is different considering most are located in research institutions, Botanical Gardens, or Museums, in opposition to most Herbaria from South America, that are housed in universities, where the curator is also a teacher, advisor in postgraduate courses and carries out administrative activities. In my point of view, the results of Table 1 bias the real scenario.
Finally, I suggest the author again brings an example for the “FAIR herbaria data through partnerships with other biological collections and datasets in the development of an extended network of biological data”. The expansion of this proposal looks innovative, but I don’t understand how can work in practice. A hypothetical example would be very welcome to clarify how the project could be implemented around the world.
Author Response
Review 2:
This review made several important comments. The reviewer requests more information about the herbaria for which no recent information is available. The reviewer requests this information on a world map, but to provide more information about individual herbaria, I have decided instead to include these data in a spreadsheet which I will submit as supplementary documentation for this article.
The Reviewer seems to want the manuscript to include more specific information about how specimen repatriation might be accomplished. However, I feel that including such details is both premature and somewhat presumptuous. If repatriation of specimens is a viable idea, then the details of how it would work would be up to the individual institutions and countries to determine. I state this in the manuscript (lines 195—197).
Reviewer 2 states some valid caveats to the data presented in Table 1. I do not agree with the assessment that the table is “biased” but I do agree that the caveats regarding the interpretation of these data should be stated more clearly, and I have added text to this effect.
Reviewer 2 also asks for an example of the extended specimen concept. The various papers cited in the text give such examples, but I do include a hypothetical example here, for clarity.
Reviewer 3 Report
Comments and Suggestions for Authors
I believe that the reviewed text is an excellent summary of the directions of development of today's herbaria and is needed to initiate a broad international discussion on the consolidation of efforts to preserve the priceless collections and use their potential in multidisciplinary research.
In the attached word file, I provide some of my thoughts that came to me after reading this manuscript. Please consider whether the data in Table 1 are definitely complete. I believe that some herbarium curators do not provide complete information for Index Herbariorum.

Author Response
Review 3:
Line 199 – The comments by Reviewer 3 here seem to be mostly this person’s personal response to the idea of repatriation, rather than suggestions for improvement of manuscript. I believe that the manuscript already addresses the situation of newly collected specimens in lines 188-190.
Table 1: The comments made here are similar to those made by Reviewer 2, and I have tried to point out the limitations of these data.
Line 290 ; Reviewer 3 wants the inclusion of a discussion of the need to invest in the proper identification. Although of course I support the proper identification, I do not agree that an emphasis on identification of already collected specimens is one of the main efforts that will sustain herbaria in the 21st century. The suggestions made in this paper for the development of broader collaborations to sustain herbaria all call for the greater use of herbarium specimens for research, and it is through such use that errors in identification will most likely be corrected.